# A Simulation Study on the Flow Behavior of Liquid Steel in Tundish with Annular Argon Blowing in the Upper Nozzle

**Xufeng Qin** [1,2]**, Changgui Cheng** [1,2,]*****, Yang Li** [1,2] **, Chunming Zhang** [1,2]**, Jinlei Zhang** [1,2] **and Yan Jin** [1,2]

[1] The State Key Laboratory of Refractories and Metallurgy, Wuhan University of Science and Technology, Wuhan 430081, China; qinxufeng@wust.edu.cn (X.Q.); liyang@wust.edu.cn (Y.L.); Zhangspringming@hotmail.com (C.Z.); Jinleiazhang@outlook.com (J.Z.); jinyan@wust.edu.cn (Y.J.)

[2] Hubei Provincial Key Laboratory for New Processes of Ironmaking and Steelmaking, Wuhan University of Science and Technology, Wuhan 430081, China

***** Correspondence: ccghlx@wust.edu.cn; Tel.: +86-027-68862651

**Abstract:** A three-dimensional mathematical model of gas−liquid two-phase flow has been established to study the flow behavior of liquid steel in the tundish. The effect of the argon flow rate and casting speed on the flow behavior of liquid steel, as well as the migration behavior of argon bubbles, was investigated. The results from the mathematical model were found to be consistent with those from the tundish water model. There were some swirl flows around the stopper when the annular argon blowing process was adopted; the flow of liquid steel near the liquid surface was active around the stopper. With increased argon flow rate, the vortex range and intensity around the stopper gradually increased, and the vertical flow velocity of the liquid steel in the vicinity of the stopper increased; the argon volume flow in the tundish and mold all increased. With increased casting speed, the vortex range and intensity around the stopper gradually decreased, the peak value of vertical flow velocity of liquid steel at the vicinity of the stopper decreased, and the distribution and ratio of argon volume flow between the tundish and the mold decreased. To avoid slag entrapment and purify the liquid steel, the argon flow rate should not be more than 3 L·min$^{-1}$. These results provide a theoretical basis to optimize the parameters of the annular argon blowing at the upper nozzle and improve the slab quality.

**Keywords:** annular argon blowing; upper nozzle; flow behavior; argon gas distribution; tundish

## 1. Introduction

The tundish is a transitional container connecting the ladle and mold. The tundish makes the liquid steel composition and temperature uniform. It also distributes liquid steel and, more importantly, facilitates the removal of inclusions and then purifies the liquid steel. Many techniques have been adopted in the tundish to remove inclusions such as retaining walls and dams, diversion walls, tundish filtering, and electromagnetic stirring [1–4]. Moreover, the argon blowing in the tundish could effectively reduce the amount of the inclusion and purify the liquid steel. The mechanism is the injection of argon gas into the liquid steel in the tundish to form bubbles. Non-metallic inclusions could then be transported to the liquid surface in the tundish for removal.

The argon blowing patterns in the tundish mainly include the long shroud, the bottom permeable brick, the stopper, and the upper nozzle. Studies [5–7] of argon blowing through the long shroud demonstrated that the micro-bubbles generated by argon blowing could improve the removal rate of

inclusions in the tundish, but these micro-bubbles had a short residence time in the tundish. Thus, the effect of removing small inclusions is less obvious.

Argon blowing through the bottom-permeable brick could effectively improve the flow pattern of liquid steel and promote the flotation and removal of fine inclusions [8–13]. The movement route of the liquid steel was more tortuous and closer to the liquid surface. This prolonged the residence time of liquid steel, improved the mixing degree of the liquid steel, and reduced the dead zone volume in the tundish. Argon blowing on the bottom-permeable brick could only stir and clean the liquid steel above the permeable brick, but this could not effectively clean the liquid steel passing through the gap between the tundish slope wall and the gas curtain. Most argon bubbles entered the tundish upper nozzle and the submerged entry nozzle (SEN) when the argon was blown by the stopper [14]. This step could clean the inner wall of the nozzle and reduce the adhesion of inclusions on the inner face of the nozzle. Moreover, some of the larger argon bubbles floated directly in the tundish and interacted with inclusions in the liquid steel during the floating process. This helped reduce the content of the inclusions in the strand [15].

The bubbles generated by the dispersed permeable portion of the upper nozzle could form a stable and continuous argon gas film between the inner wall of the nozzle and liquid steel [16–18]. This could effectively suppress the accumulation of inclusions such as $Al_2O_3$ on the inner wall of the nozzle and reduce the risk of nozzle clogging. Concurrently, argon bubbles generated from the ruptured gas film could wash the inclusions deposited on the inner wall of the nozzle [19].

Smirnov et al. [20–23] studied the argon-blowing process through a gas-permeable ceramic rod embedded in the nozzle pocket brick. Here, the argon bubbles raised around the stopper and formed an annular gas curtain barrier in the tundish. Their work showed that the technique could reduce the adhesion of inclusions on the inner face of the nozzle and prevent the nozzle from clogging. The loss of refractory material was significantly reduced versus argon blowing on the bottom-permeable brick. However, the void region between the permeable ceramic rods could make the liquid steel entering the tundish nozzle insufficiently clean. This would weaken the ability to remove inclusions in the liquid steel.

The preceding studies are significant for effectively controlling the liquid steel flow by argon blowing—this purifies the liquid steel in the tundish and prevents the nozzle from clogging. Here, we propose to use an annular permeable brick with a certain width set in a pocket brick on the outside of the upper nozzle to form a relatively complete annular gas curtain around the stopper. This is based on the work of Smirnov et al. and will improve the effect of controlling fluid flow in the tundish. The rising argon bubbles may promote the removal of inclusions in liquid steel of the tundish. Concurrently, the argon bubbles partially entering into the nozzle can realize the function of argon blowing with the stopper or upper nozzle to prevent the nozzle from clogging.

In this paper, we describe a three-dimensional mathematical model for the annular argon blowing at the upper nozzle in the tundish based on the actual process conditions of a continuous slab-casting tundish in a steel plant. The discrete phase model (DPM) was used to simulate the argon blowing process to analyze the effects of different argon flow rates and casting speed on the flow behavior of liquid steel and the migration behavior of argon bubbles in the tundish. In addition, the flow behavior of liquid steel with the annular argon blowing at upper nozzle was analyzed and compared with the established water model of the tundish. The results can be leveraged as a theoretical basis for the optimization of the annular argon blowing at the upper nozzle and the improvement of slab quality.

## 2. Model Description

### 2.1. Model Assumption

(1) The effect of liquid slag on the flow behavior of liquid steel in the tundish is neglected.

(2) The flow of liquid steel is a transient incompressible flow, and the physical properties of the liquid steel such as the density and viscosity are constant.

(3) Argon bubbles are regarded as rigid spheres, the bubble size does not change during the ascent, and the bubble diameters are distributed by Rosin−Rammler statistics, which were obtained by the water model experiments of tundish.

(4) The transport of tracer in the tundish is an unsteady mass transfer process.

*2.2. Governing Equations*

The flow of the liquid steel in the tundish is a three-dimensional transient incompressible flow and mass transfer process, which satisfies the basic physical laws of mass, momentum conservation. The continuity equation and the momentum equation are described as follows.

Continuity equation:

$$\frac{\partial \rho}{\partial t} + \frac{\partial (\rho u_i)}{\partial x_i} = 0 \tag{1}$$

Momentum equation (N–S):

$$\frac{\partial (\rho u_i)}{\partial t} + \frac{\partial (\rho u_i u_j)}{\partial x_j} = -\frac{\partial P}{\partial x_i} + \frac{\partial}{\partial x_j}\left[\mu_{\text{eff}}\left(\frac{\partial u_i}{\partial x_j} + \frac{\partial u_j}{\partial x_i}\right)\right] + \rho g + F_g \tag{2}$$

where $\rho$ is the fluid density, in $\text{kg·m}^{-3}$; $u_i$ and $u_j$ are the velocity vectors, in $\text{m·s}^{-1}$, $i$ and $j$ each represent the three coordinate directions ($x$, $y$, and $z$), and repeated indices imply summation; $P$ is the pressure, in Pa; $\mu_{\text{eff}}$ is the turbulent effective viscosity coefficient, in Pa·s; g is the gravitational acceleration, in $\text{m·s}^{-2}$. $F_g$ is a momentum source term, which accounts for the presence of argon bubbles, in $\text{N·m}^{-3}$. Here, the standard *k-ε* turbulence equations were used in the mathematical model. The governing equations describing turbulent kinetic energy ($k$) and the dissipation rate of turbulence energy ($\varepsilon$) are, respectively:

$$\frac{\partial}{\partial t}(\rho k) + \frac{\partial}{\partial x_i}\left(\rho u_i k - \frac{\mu_{\text{eff}}}{\sigma_k}\frac{\partial k}{\partial x_i}\right) = G - \rho\varepsilon \tag{3}$$

$$\frac{\partial}{\partial t}(\rho\varepsilon) + \frac{\partial}{\partial x_i}\left(\rho u_i\varepsilon - \frac{\mu_{\text{eff}}}{\sigma_\varepsilon}\frac{\partial k}{\partial x_i}\right) = \frac{1}{k}\left(C_1 G - C_2\rho\varepsilon^2\right) \tag{4}$$

$$G = \mu_t\frac{\partial u_i}{\partial x_i}\left(\frac{\partial u_i}{\partial x_i} + \frac{\partial u_i}{\partial x_j}\right) \tag{5}$$

$$\mu_{\text{eff}} = \mu_0 + \mu_t = \mu_0 + \rho C_\mu\frac{k^2}{\varepsilon} \tag{6}$$

where $\mu_0$ is the dynamic viscosity, in Pa·s; $\mu_t$ is the turbulent viscosity, in Pa·s; $k$ is the turbulent kinetic energy of the fluid, in $\text{m}^2\text{·s}^{-2}$; $\varepsilon$ is the turbulent energy dissipation rate, in $\text{m}^2\text{·s}^{-3}$. Terms $C_1$, $C_2$, $C_\mu$, $\sigma_k$, and $\sigma_\varepsilon$ are empirical constants. The recommended values [11] of Launder and Spalding are $C_1$ = 1.42, $C_2$ = 1.92, $C_\mu$ = 0.09, $\sigma_k$ = 1.0, and $\sigma_\varepsilon$ = 1.0.

The trajectories and distributions of the argon bubbles are simulated using the discrete phase model (DPM). An equation for argon bubble velocity is obtained considering the drag force, buoyancy force, and virtual mass force exerted by the fluid on bubbles:

$$\frac{du_g}{dt} = \frac{18\mu}{\rho_g d_g^2}\cdot\frac{C_D \text{Re}_g}{24}(u_i - u_g) + \frac{\pi d_g^3}{6}(\rho_g - \rho)g + \frac{1}{2}\frac{\rho}{\rho_g}\frac{d}{dt}(u_i - u_g) \tag{7}$$

where $u_g$ is the bubble velocity, in $\text{m·s}^{-1}$; $\mu$ is the molecular viscosity of the fluid, in Pa·s; $\rho_g$ is the argon density, in $\text{kg·m}^{-3}$; $d_g$ is the bubble diameter, in m; $\text{Re}_g$ is the relative Reynolds number of bubbles; and $C_D$ is the drag coefficient [24], which is a function of $\text{Re}_g$:

$$C_D = \begin{cases} \frac{24}{\text{Re}_g}\left(1 + \frac{1}{6}\text{Re}_g^{\frac{2}{3}}\right) & \text{if} \text{Re}_g < 1000 \\ 0.44 & \text{if} \text{Re}_g \geq 1000 \end{cases} \tag{8}$$

$$\text{Re}_g = \frac{\rho d_g |u_g - u_i|}{\mu} \tag{9}$$

The momentum transfer from the discrete phase towards the melt is computed by examining their momentum change as [11,25]:

$$F_g = \sum_j^N \left( \frac{3\mu_0 C_D Re_g}{4\rho_g d_g^2} (u_{gj} - u_i) \right) m_p \Delta t \tag{10}$$

where $N$ is the number of bubbles in a computational cell, which can be determined by the particle trajectory unsteady tracking method in Fluent software; $u_{gj}$ is the velocity of bubble in a computational cell, in $m \cdot s^{-1}$, $m_p$ is the mass flow rate of argon bubbles, in $kg \cdot s^{-1}$, which equals the argon density multiplied by the argon flow rate; $\Delta t$ is the time step, in s, its value is 0.005 s.

In Equation (7), $d_g$ adopts a Rosin-Rammler distribution, and the different bubble size range is divided into discrete size groups as shown in Equation (11).

$$Y_d = e^{-(d_g/\bar{d})^n} \tag{11}$$

where $Y_d$ is the mass fraction with bubble diameter greater than $d_g$; $\bar{d}$ is the average diameter of bubbles, in m; and $n$ is the distribution index. The mass flow with bubble diameter greater than $d_g$ equals the mass flow rate of argon bubbles multiplied by the time step and the mass fraction with bubble diameter greater than $d_g$, the mass flow of different diameter range can be determined, and then the number of argon bubbles of different diameter range entering into tundish from the annular argon blowing brick can be determined.

In order to determine the residence time of liquid steel [26], the time evolution of tracer concentration C in the tundish was described by Equation (12):

$$\frac{\partial}{\partial t}(\rho C) + \frac{\partial}{\partial x_i}(\rho u_i C) = \frac{\partial}{\partial x_i}\left(\rho D_{eff} \frac{\partial C}{\partial x_i}\right) \tag{12}$$

$$D_{eff} = D_0 + \frac{\mu_{eff}}{\rho Sc_t} \tag{13}$$

where C is the concentration of the tracer, in $kg \cdot m^3$; $D_{eff}$ is the effective diffusion coefficient, in $m^2 \cdot s^{-1}$; $D_0$ is the molecular diffusion coefficient in $m^2 \cdot s^{-1}$, and its value is 0; and $Sc_t$ is the turbulent Schmidt number and its value is 0.7.

*2.3. Boundary Conditions*

(1) The model inlet of tundish was set as the velocity-inlet, and the entry velocity was calculated based on the mass conservation principle according to the section size of the strand, the casting speed, and the inner diameter of the long shroud, namely, the velocity-inlet is equal to that the cross-sectional area of strand is multiplied by the casting speed and divided by the cross section area of the long shroud. The inlet value of $k$, the turbulence kinetic energy, and the $\varepsilon$, the rate of turbulence energy dissipation, were estimated from the following relations:

$$k = 0.01 u_{in}^2 \tag{14}$$

$$\varepsilon = \frac{k^{1.5}}{0.5 D_{in}} \tag{15}$$

where $u_{in}$ is the inlet velocity, in $m \cdot s^{-1}$, and $D_{in}$ is the diameter of the inlet, in m. The value of $D_{in}$ is 0.07 m. When the casting speed was controlled to be 1.05 $m \cdot min^{-1}$, 1.2 $m \cdot min^{-1}$, 1.35 $m \cdot min^{-1}$ and 1.5 $m \cdot min^{-1}$, the value of $u_{in}$ was 1.965 $m \cdot s^{-1}$, 2.246 $m \cdot s^{-1}$, 2.527 $m \cdot s^{-1}$ and 2.808 $m \cdot s^{-1}$, respectively.

(2) The outlet of the liquid steel was set as the pressure-outlet according to the immersion depth of the SEN; the model outlet was the escape outlet for the argon bubbles.

(3) The surface of the molten pool was set as a free-surface [27], the normal velocity component and normal gradients of all other variables were assumed to be zero, and the bubbles were trapped at the liquid surface.

(4) Owing to the bilateral symmetry, only a half of the tundish is considered in the calculation in order to lower the computation cost. On symmetry plane, the boundary condition for velocity field is a zero normal component and zero gradient of tangential velocity component.

(5) The wall of the tundish was modeled as a no-slip wall boundary condition; the region near the wall was treated with a standard wall function [28].

### 2.4. Numerical Method

A three-dimensional mathematical model was established according to the real size of the industrial tundish. The schematic diagram of the tundish vertical view is shown in Figure 1. The half-tundish was taken as the computational domain considering the symmetry of the tundish. A schematic diagram of annular argon blowing at the upper nozzle in the tundish is shown in Figure 2. The physical properties and operational parameters are shown in Table 1.

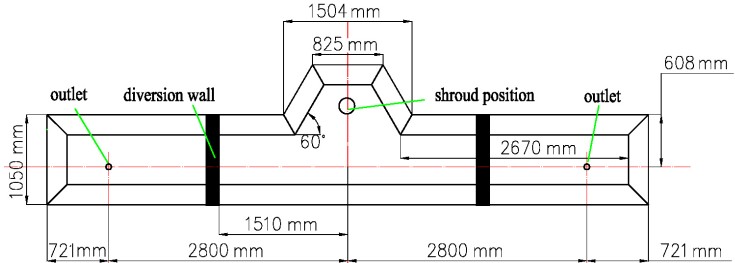

**Figure 1.** Schematic diagram of tundish vertical view.

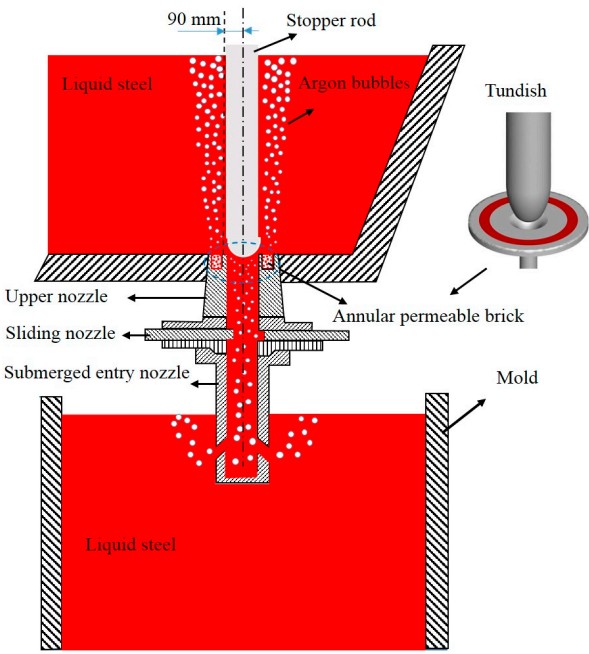

**Figure 2.** Schematic diagram of annular argon blowing at the tundish upper nozzle.

**Table 1.** Physical properties and process parameters.

| Parameters | Value |
|---|---|
| Liquid steel density (kg·m$^{-3}$) | 7000 |
| Liquid steel viscosity (Pa·s) | 0.0065 |
| Argon gas density (kg·m$^{-3}$) | 0.27 |
| Working liquid surface height of tundish (mm) | 960 |
| Inner and outer diameter of the shroud (mm) | 70/120 |
| Immersion depth of the shroud (mm) | 250 |
| Diameter of the stopper (mm) | 127 |
| Inner diameter of the upper nozzle (mm) | 50 |
| Sectional dimensions of the slab (mm × mm) | 1235 × 175 |

The coordinate axis X was parallel to the intersection line of the tundish front wall and the bottom wall, and the Y direction was perpendicular to the X direction. A schematic diagram of the computational domain coordinate system is shown in Figure 3, and the flow field in XZ plane and YZ plane will be shown for analyzing the effect of the argon bubbles on the flow of liquid steel.

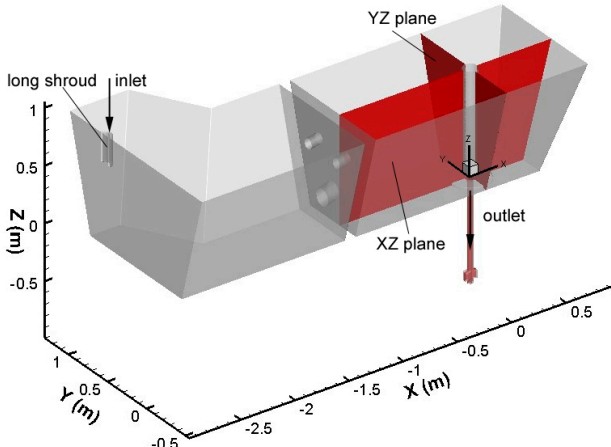

**Figure 3.** Schematic diagram of regional coordinate system of half tundish in the model calculation.

A mathematical model was developed using the finite-volume-based program ANSYS-FLUENT software (16.0, Ansys Inc., Canonsburg, PA, USA) based on these assumptions including the governing equations and boundary conditions described above. The velocity of liquid steel at the inlet and outlet of tundish is larger, and the flow behavior in the vicinity of stopper should be paid more attention to. So, local grid refinement was applied to simulate the behavior of blowing argon in the tundish; the meshes of the FLUENT computational domain included 6,700,000 unstructured grids using ANSYS-ICEM software (16.0, Ansys Inc.). The mesh size is 2 mm in the argon bubble action zone, and 8 mm far from the stopper. The SIMPLEC algorithm was applied to the velocity−pressure coupling. The second upwind order scheme was employed for discretization of momentum, *k*, and *ε* equations. The calculation was considered to be converged when the normalized residuals of all variables were smaller than $10^{-4}$, when the flow field could achieve a relatively stable state. To reduce the simulation time and ensure the balance of equations at a discrete point of time, a fixed time step of 0.005 s was used in the time-dependent solution.

Considering the interaction between the continuous phase and the discrete phase, the stable flow field of the argon blowing in the tundish was obtained by calculating 15 s. Based on the stable flow field of gas–liquid two-phase, the tracer with the same properties as the liquid steel was added to the inlet of the stable flow field of the tundish for 1 s. The mixed flow of the tracer and the regional fluid were calculated by the transient mode simulation for 3000 s to obtain the residence time distribution (RTD) curve. The flow characteristics of the liquid steel in the tundish under different conditions were

obtained by analyzing the RTD curve using the modified model proposed by Hong [29]. Next, we compared the vertical velocity of the liquid steel in the vicinity of stopper to analyze the influence of different process parameters on the flow of liquid steel. This was done 90 mm from the center of the stopper. The argon gas distribution between the tundish and mold was obtained by counting the number of bubbles entering the nozzle and the tundish in one second. The experimental outline is shown in Table 2. Combined with the results of water model experiments of tundish, the parameters of Rosin–Rammler distribution used in the calculation is shown in Table 3, then the mass fraction with bubble diameter greater than $d_g$ can be determined by using of Equation (11).

**Table 2.** Mathematical simulation scheme.

| Case | Casting Speed (m·min$^{-1}$) | Argon Flow Rate (L·min$^{-1}$) | Inner and Outer Diameter of Annular Permeable Brick (mm) |
|---|---|---|---|
| Case 1 | 1.35 | 2, 3, 4, 5 | 220/280 |
| Case 2 | 1.05, 1.20, 1.35,1.50 | 3 | 220/280 |

**Table 3.** Parameters of Rosin–Rammler distribution used in calculation.

| Argon Flow Rate (L·min$^{-1}$) | Minimum Bubble Diameter (mm) | Maximum Bubble Diameter (mm) | Average Bubble Diameter (mm) | Distribution Index |
|---|---|---|---|---|
| 2 | 0.6 | 2.85 | 1.6 | 2.82 |
| 3 | 0.65 | 2.90 | 1.8 | 4.61 |
| 4 | 0.7 | 2.95 | 2.0 | 5.08 |
| 5 | 0.80 | 3.0 | 2.2 | 7.96 |

## 3. Comparison of Flow Behavior in Mathematical Model and Water Model

The flow of liquid steel in the tundish is mainly affected by the viscous force, gravity, and inertial force. The Froude number was chosen to ensure the motion similarity between the prototype and the model. A water model of the tundish with a 1:2 scale was made to simulate the argon blowing through the annular permeable brick in this work. When the argon flow rate was 3 L·min$^{-1}$ and the casting speed was 1.35 m·min$^{-1}$, the inner and the outer diameters of the annular permeable brick were 220 mm and 280 mm, respectively. The distribution of argon bubbles in the water model and numerical simulation is shown in Figure 4, and the diffusion of the tracer in the tundish at the different times is shown in Figure 5, the left-hand diagram in Figure 5 shows the calculated flow of liquid steel in cross-section which is through the center of the lower diversion hole and the right side of the retaining wall.

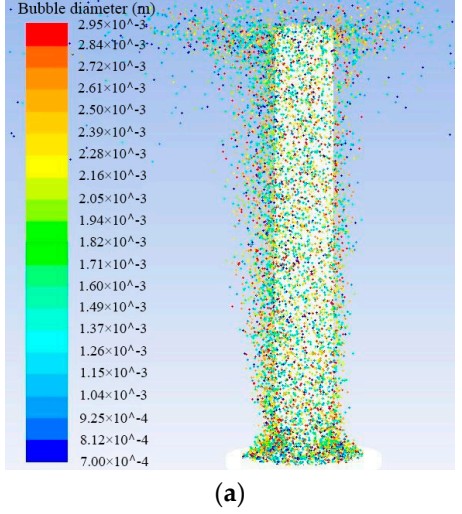

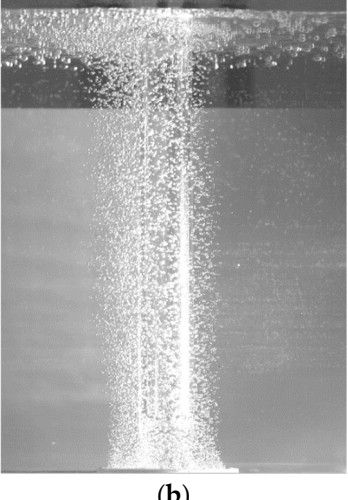

(a)  (b)

**Figure 4.** Distribution of argon bubbles in the tundish: (**a**) numerical simulation and (**b**) water model.

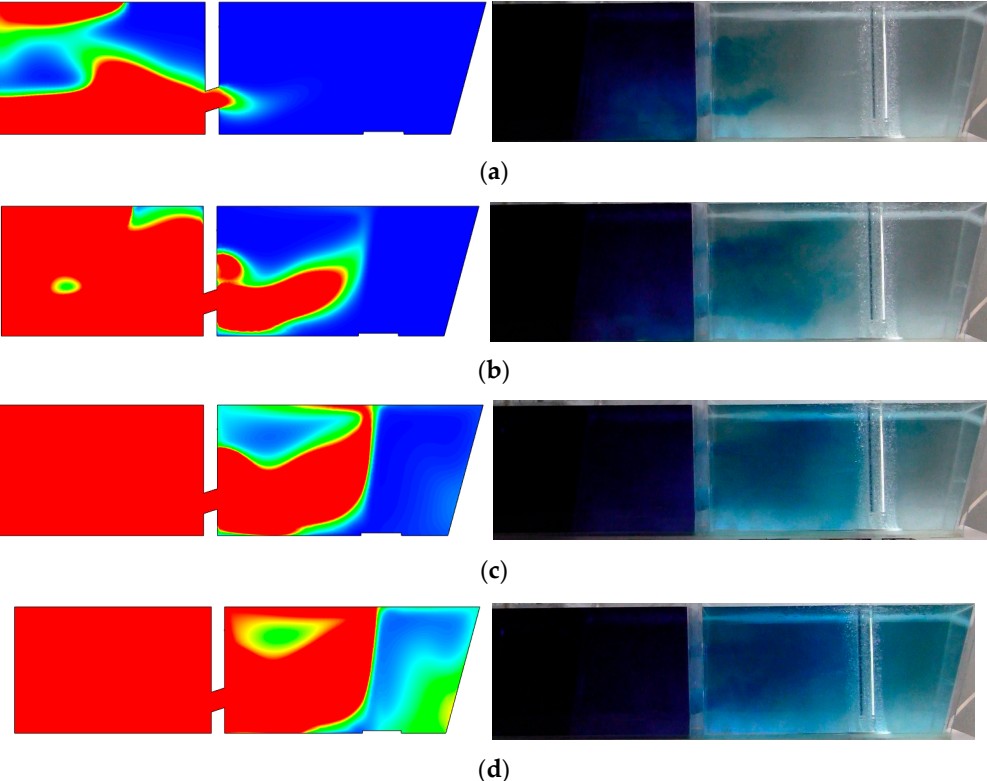

**Figure 5.** Flow field of the tracer in the half-tundish at different times of numerical model (left-hand diagram) and experiment (right-hand diagram): (**a**) 10 s, (**b**) 20 s, (**c**) 30 s, and (**d**) 40 s.

Figure 4 shows that the argon bubbles are asymmetrical on both sides of the stopper. The number of argon bubbles on the left side of the stopper is significantly higher than that on the right side of the stopper due to the action of the liquid steel flow.

Figure 5 demonstrates that the liquid steel departs from the diversion hole at the vicinity of the stopper and then flows upward to the liquid surface with the floating argon bubbles. The left side in Figure 5 is the flow field obtained by the mathematical modelling, right side is the flow field in water model of tundish. There are multi-swirl flow zones around the stopper. The diversion hole is not in the same XZ plane. It is part of the liquid steel that migrates from the back area of the stopper to the right wall of the tundish. There is weakened swirl flow to the right of the stopper. The distribution behavior of the argon bubbles calculated by the mathematical model is similar to that in the water model. Thus, the flow field of the liquid steel calculated by the numerical simulation is similar to that in the water model experiment.

## 4. Results and Discussion

### 4.1. Typical Flow Behavior of Liquid Steel in the Tundish with Annular Argon Blowing in the Upper Nozzle

When the casting speed was controlled at 1.35 m·min$^{-1}$, the inner and outer diameter of the annular permeable brick were 220 mm and 280 mm, respectively. The velocity streamlines of liquid steel in the tundish without the argon blowing and with the argon blowing at 3 L·min$^{-1}$ are shown in Figure 6.

Figure 6a demonstrates that the liquid steel entered into the tundish from the shroud, and spreads after impinging on the turbulence controller. The liquid steel then passes through the diversion holes of the retaining wall and migrates obliquely upward. For the viscous resistance of the liquid steel and the obstruction of the stopper, the velocity at the near-surface of the liquid steel around the stopper is reduced gradually when the argon blowing is not adopted. Thus, the laminar flow in the liquid surface is weak, and there is no rising flow around the stopper.

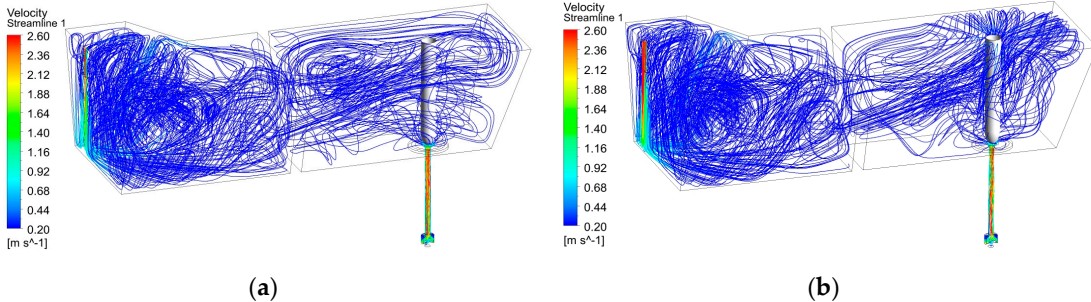

**Figure 6.** Velocity streamline of the liquid steel in the tundish: (**a**) without argon blowing and (**b**) with argon blowing.

When argon is blown through the annular permeable brick, the streamline of the liquid steel in the pouring area is obviously changed versus the pouring area without argon. Figure 6b demonstrates that the liquid steel is driven by the floating bubbles. It moves up to the liquid surface and diffuses to the surrounding area of the stopper. The liquid steel then moves to the vicinity of the tundish wall and flows downward; some swirl flows near the stopper are formed, which can make the level flow of liquid steel around the stopper active. The flow path of the liquid steel is extended, and the short circuit flow is significantly decreased in the tundish.

The flow characteristics of the liquid steel in the tundish with and without argon blowing by the mathematical calculation are shown in Table 4. Table 4 shows that the average residence time and the volume fraction of plug flow of liquid steel in the tundish increase with blowing argon versus no argon blowing. Thus, the annular argon blowing is beneficial to the floating and removal of inclusions in liquid steel.

**Table 4.** Flow characteristics of the liquid steel in the tundish with and without blowing argon.

| Process Condition | Average Residence Time (s) | Volume Fraction of Plug Flow (%) | Volume Fraction of Dead Zone (%) | Volume Fraction of Mixed Flow (%) |
|---|---|---|---|---|
| without argon | 592.12 | 30.91 | 12.75 | 56.34 |
| with argon | 593.56 | 31.59 | 12.53 | 55.88 |

*4.2. Effect of Flow Rate of the Argon Blowing on the Flow Behavior of Liquid Steel in the Tundish*

Figure 7 shows the flow behavior of the liquid steel along the XZ plane and the XY plane around the stopper in the tundish, the width of the XZ plane is only the distance between the diversion wall centerline and the right side wall of the tundish. The casting speed was 1.2 m·min$^{-1}$, and the inner and outer diameters of the annular permeable area in the nozzle pocket brick were 220 mm and 280 mm, respectively. The argon flow rate was 2 L·min$^{-1}$, 3 L·min$^{-1}$, 4 L·min$^{-1}$, and 5 L·min$^{-1}$. Under the same conditions, the vertical velocity of the liquid steel in the vicinity of the stopper is shown in Figure 8. The argon flow rate of 0 in Figure 8 indicates that the argon flow was absent.

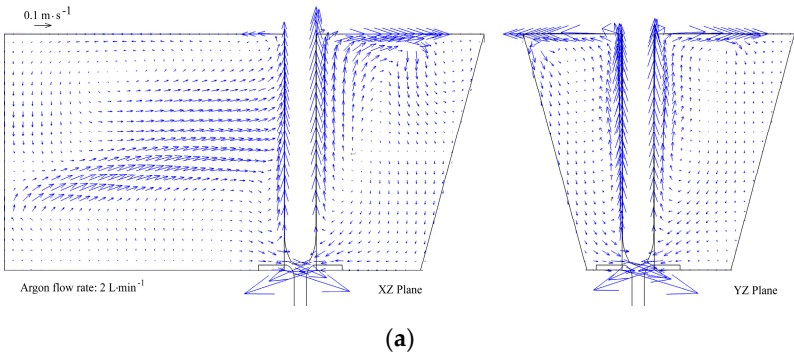

**Figure 7.** *Cont.*

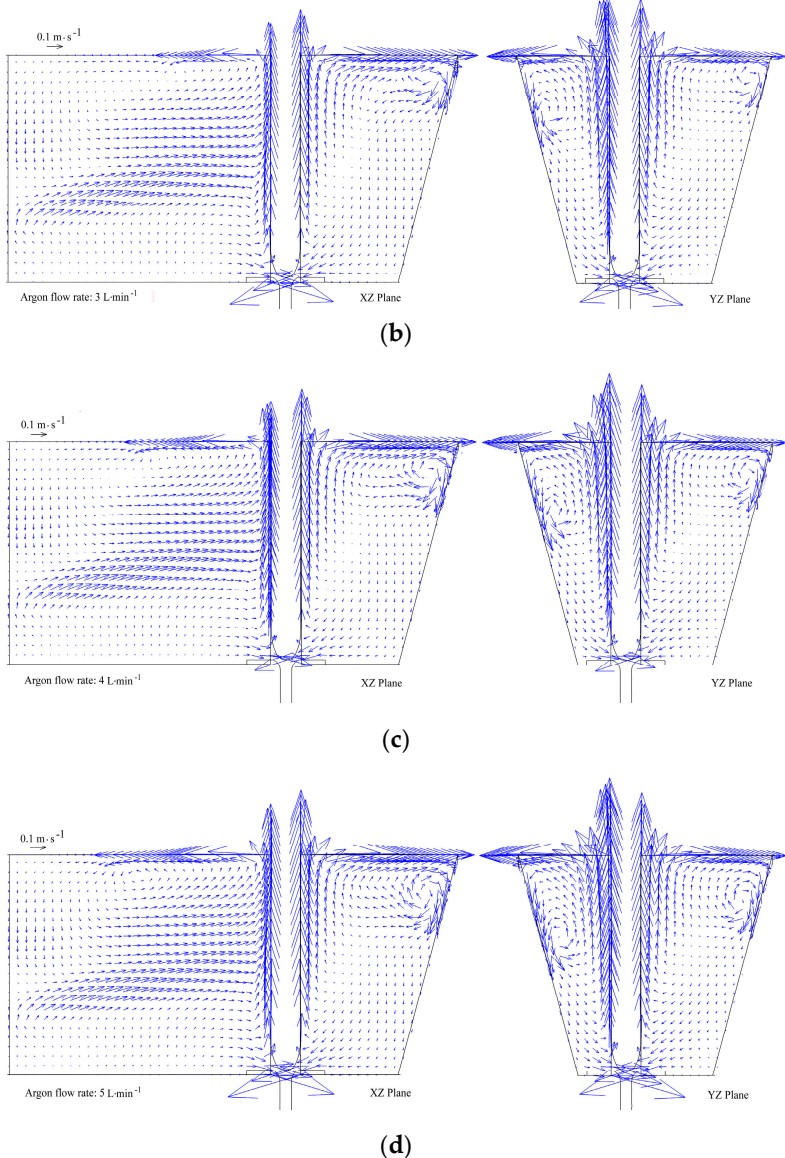

**Figure 7.** Flow behavior of liquid steel in the tundish under different argon flow rates: (**a**) 2 L·min$^{-1}$, (**b**) 3 L·min$^{-1}$, (**c**) 4 L·min$^{-1}$, and (**d**) 5 L·min$^{-1}$.

The vortex range and intensity around the stopper increase gradually with increased argon flowrates (Figure 7). The vortex center of liquid steel in the front, rear, and right sides of the stopper move towards the tundish wall, and then move downward along the tundish wall. The vortex center of liquid steel in the left side of the stopper moves towards the diversion wall. These flow behaviors are related to the coupling effect of the upflow of liquid steel near the stopper and the normal flow from the diversion hole. The flow intensity of the liquid steel near the tundish wall and diversion wall then increase, which decreases the dead zone of liquid steel in the tundish.

The near surface flow of liquid steel can promote floating of inclusions in the tundish, and then the increased flow rate of argon may improve the purity of liquid steel. However, an excessive surface flow velocity of liquid steel may cause slag entrapment. When the argon flowrate was 2 L·min$^{-1}$, 3 L·min$^{-1}$, 4 L·min$^{-1}$, and 5 L·min$^{-1}$, the peak value of the surface flow velocity in the tundish was 0.38 m·s$^{-1}$, 0.48 m·s$^{-1}$, 0.52 m·s$^{-1}$, 0.57 m·s$^{-1}$, respectively. Slag entrapment may occur at surface flow velocities over 0.45 m·s$^{-1}$ [30]. The argon flow rate should therefore not be larger than 3 L·min$^{-1}$.

As the distance from the upper nozzle surface increases, Figure 8 demonstrates that the vertical flow velocity of the liquid steel in the vicinity of the stopper first increases and then decreases.

It approaches 0 at the liquid surface in the tundish with blowing argon. These are related to the vortex flow around the stopper when the liquid steel arrives at this level. The vertical flow velocity is 0 while the horizontal flow velocity of liquid steel is the largest. In the mathematical model, the liquid surface was set to the free liquid surface in which the velocity variable at the normal direction is zero. In practical operation, the liquid surface of the tundish is covered with a slag layer. The floating argon bubbles around the stopper drive the liquid steel upward. The liquid steel from the left diversion hole impinges on the back side of the stopper, and some argon bubbles were carried by the liquid steel to the right side of the stopper as the argon bubbles floated. The vertical flow velocity is then higher than that in the left-hand of the stopper. Moreover, the distance between the front wall of the tundish and stopper is smaller, the floating argon bubbles cannot easily be dispersed. The driving effect of the floating bubbles then increases, and the vertical flow velocity of the liquid steel in the front vicinity of the stopper is higher.

There is no upflow of liquid steel without argon blowing. The liquid steel in the vicinity of the stopper flowed directly into the upper nozzle, and the vertical flow velocity in Figure 8 is negative. With increased argon flow rate, the driving effect of argon bubbles increases, and the vertical flow velocity of the liquid steel in the vicinity of the stopper increases. The peak value of vertical flow velocity of liquid steel increases with increased argon flow rate, and the peak value position also increases. The difference in peak value and peak value position with vertical flow velocity in different orientations is related to the coupling effect of the upflow of liquid steel near the stopper and the normal flow from the diversion hole.

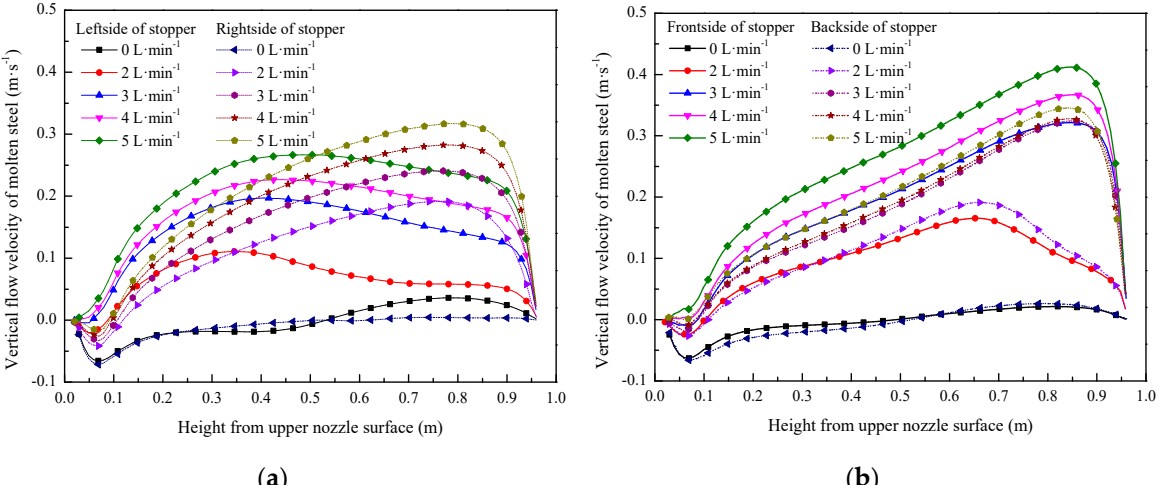

**Figure 8.** Vertical flow velocity of liquid steel in the vicinity of the stopper under different flow rates of argon: (**a**) left side and right side of the stopper; (**b**) front side and back sides of the stopper.

Figure 9 shows the volume flow distribution and ratio of argon in the tundish and mold under different argon flow rates. With increased argon flow rates, more argon bubbles may be released from the annular permeable brick. The number of argon bubbles entering the tundish and SEN then increases. The argon bubbles entering into the tundish nozzle can prevent the nozzle from clogging, and the high flow rate of argon may be appropriate. The bubble dimension increases with increasing argon flow rate. The larger bubbles float more easily, and the ratio of volume flow of argon between the tundish and mold then increases with increasing argon flow rate.

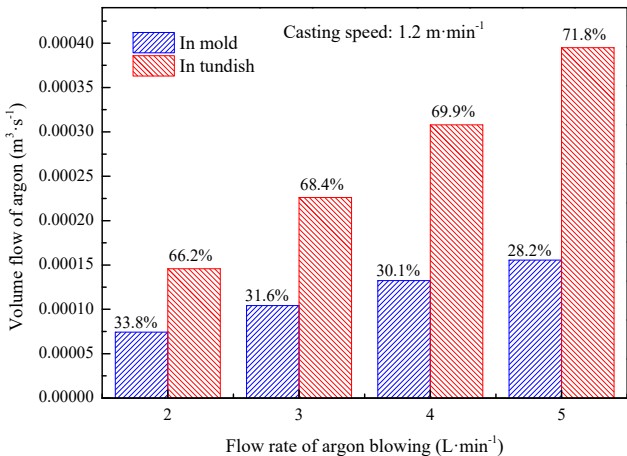

**Figure 9.** Volume flow distribution of argon under different argon flow rate.

*4.3. Effect of Casting Speed on the Flow Behavior of Liquid Steel in the Tundish*

When the flow rate of argon blowing was 3 L·min$^{-1}$, the inner and outer diameter of the annular permeable area in nozzle pocket brick were fixed at 220 mm and 280 mm, respectively, and the casting speed was 1.05 m·min$^{-1}$, 1.20 m·min$^{-1}$, 1.35 m·min$^{-1}$ and 1.50 m·min$^{-1}$. The flow behavior of the molten steel in the XZ plane and the XY plane at different casting speeds is shown in Figure 10. The vertical flow velocity of liquid steel in the vicinity of the stopper at different casting speeds is shown in Figure 11.

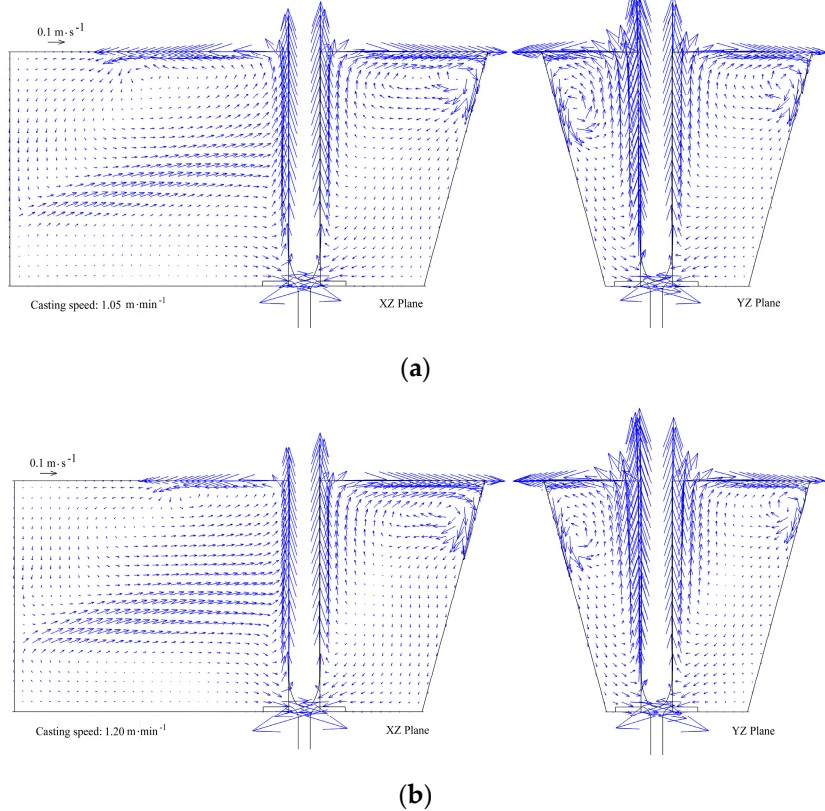

**Figure 10.** *Cont.*

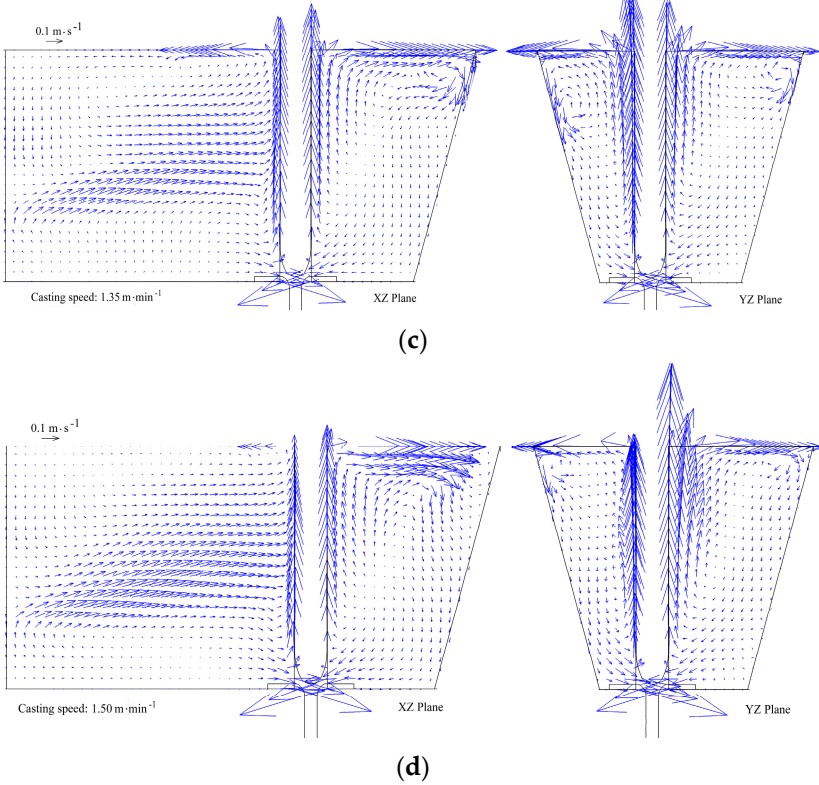

**Figure 10.** Flow behavior of liquid steel in the tundish under different casting speeds: (**a**) 1.05 m·min$^{-1}$, (**b**) 1.20 m·min$^{-1}$, (**c**) 1.35 m·min$^{-1}$, and (**d**) 1.50 m·min$^{-1}$.

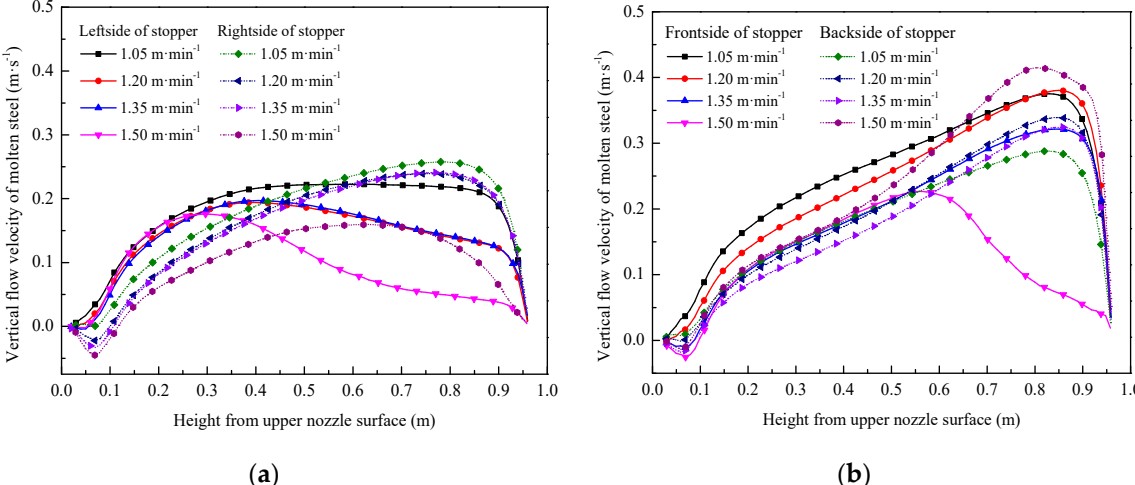

**Figure 11.** Vertical velocity of liquid steel in near area of stopper at different casting speeds: (**a**) left and right sides of the stopper; (**b**) front and back sides of the stopper.

Figure 10 shows that the vortex range and intensity around the stopper decrease gradually with increased casting speed. The vortex center of the liquid steel in different orientations moves towards the stopper. A higher casting speed leads to more bubbles entering the nozzle. The floating effect of the argon bubbles around the stopper weakens.

When the casting speed was 1.05, 1.20, 1.35, and 1.50 m·min$^{-1}$, the flow velocity peak of the liquid surface is 0.46, 0.50, 0.48, and 0.59 m·s$^{-1}$, respectively. The high casting speed can weaken the floating effect of the argon bubbles. The flow intensity near the back side of the stopper is high. This is related to the high flow velocity of liquid steel in the diversion hole; thus, the flow velocity peak at the

liquid surface is high when the casting speed is 1.50 m·min$^{-1}$. This may lead to a slag entrapment problem, which can be eliminated by increasing the diameter of the diversion hole under the high casting speeds.

The peak value of the vertical flow velocity of liquid steel decreases with increased casting speed (Figure 11). This is seen at the left, right, and front side of the stopper; the height of peak value position also decreases. While the peak value of the vertical flow velocity of liquid steel at the back side of the stopper increases, the peak value position also moves up—this is related to the high flow velocity of liquid steel in the diversion hole, and can lead to slag entrapment. Thus, a high casting speed is disadvantageous for removal of inclusions.

Figure 12 shows the volume flow distribution and ratio of argon in the tundish and mold under different casting speeds. More argon bubbles were brought into the tundish nozzle with increased casting speed. The ratio of volume flow of argon between the tundish and mold decreases. When the casting speed is controlled to 1.05 m·min$^{-1}$, the argon volume fraction in the mold is minimized accounting for 15.9% of the total volume of blowing argon. This is 38.6% when the casting speed is controlled to 1.5 m·min$^{-1}$. The lower casting speed can promote the removal of nonmetallic inclusions in the tundish for a high floating effect around the stopper. This increases the residence time of the liquid steel. The high casting speed can catch more bubbles into the nozzle, which is beneficial to prevent the nozzle from clogging. It is important to regulate the floating bubbles around the stopper and the argon volume entering the nozzle.

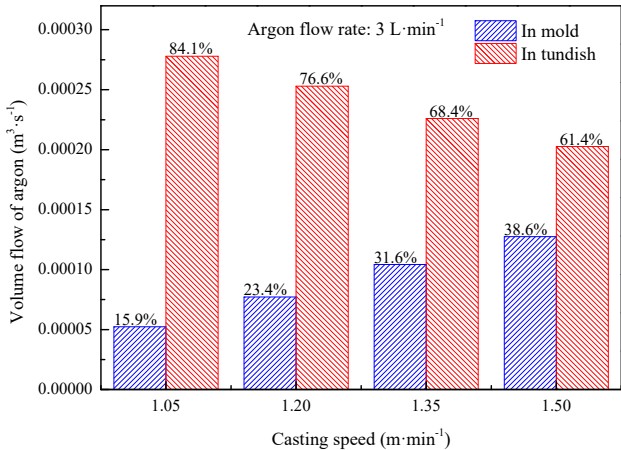

**Figure 12.** Volume flow distribution of argon under different casting speeds.

## 5. Conclusions

(1) There was some swirl flow of liquid steel around the stopper when the annular argon blowing at the upper nozzle was adopted. The liquid surface flow of liquid steel around the stopper is active, and the average residence time of liquid steel in the tundish slightly increases.

(2) With increased argon flow, the vortex range and intensity around the stopper gradually increase, and the vertical flow velocity of the liquid steel in the vicinity of the stopper increases. The argon volume flows in the tundish and the mold both increase.

(3) With increased casting speed, the vortex range and intensity around the stopper gradually decrease. The peak value of the vertical flow velocity of liquid steel at the vicinity of the stopper decreases, and the distribution and ratio of argon volume flow between the tundish and the mold decrease.

**Author Contributions:** X.Q. and C.C. conceived and designed the study; X.Q. and Y.L. conducted the experiment; X.Q. analyzed the experimental data and wrote the manuscript with the advice of C.C., C.Z., J.Z., and Y.J.

**Funding:** This research was funded by the National Nature Science Foundation of China (No. 51874215 and No. 51504172).

**Acknowledgments:** The authors would like to acknowledge the financial support from National Science Foundation of China (Grant No. 51874215 and No. 51504172).

**Conflicts of Interest:** The authors declare no conflicts of interest.

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
