# Peer review of "A Simulation Study on the Flow Behavior of Liquid Steel in Tundish with Annular Argon Blowing in the Upper Nozzle"

_metals, doi:10.3390/met9020225_

Reviewer 1 Report

The authors present a simulation study by CFD of the conditions in a tundish in steelmaking, with argon blowing. The authors simulate different argon feed rates and casting speeds and draw conclusions on whether the flow patterns are appropriate to remove inclusions and too yield desired flow patterns and residence time of the steel in the tundish. The simulated cases are clearly presented and the flow of the manuscript is good. Many results are reported, but it would be good to come up with more definite findings for the reader, i.e., to stress what can be learnt from the computational exercise. Some of the findings reported are rather obvious and could be given less emphasis (e.g., on l. 199 that molten steel entered the tundish from the shroud, and on l. 264 that at a higher flow rate of argon, more argon bubbles may be released from the annular permeable brick).  On the other hand, some quite definite findings are reported that are not justified, e.g., l. 215: Thus, the flow behavior of liquid steel in the tundish is reasonable (in what sense!?) when the annular argon blowing was adopted – the inclusions can float more easily (not demonstrated in any way in the analysis!).

The verification of the computational model is not appropriate. The authors’ section (3) on this topic merely holds simulation results and a few photographs. The bubble plume photo (Fig. 3b) does not allow for any further comparison with the simulated results, and in Fig. 4 the authors don’t even bother to say what the right figures represent. The CFD simulations in the latter figure have no scale, and to me the comparison gives practically nothing: the authors conclude that the verification indicates that the model is credible, but the material in section does not support the claim. Here, the authors have to be much more specific and also quantify their findings in some way.

The language of the paper is not bad but should still be improved, as there are numerous smaller problems. For instance, the authors use singular for bubble and inclusion where plural would be appropriate. In my detailed list below I go through the paper with minor questions/comments, reporting the line (i.) number:

Throughout the text: Replace “liquid level” by “liquid surface”

l. 12: model for  => model of

l. 15: were consistent  => were found consistent

l. 24: to avoid purity??

l. 30: Tundish metal => the tundish

l. 33: the inclusion => inclusions

l. 36: this nonmetallic inclusion => Nonmetallic inclusions

l. 43: on the => through the

l. 44: floatation => flotation

l. 46: Why mention both liquid and molten steel – are they different?

l. 48: brick. This => brick, but

l. 50: entered into => entered

l. 51: argon blowed by stopper?

l. 53: inclusion = inclusions

l. 60: et al => et al.

l. 66: the tundish nozzle unable to be sufficiently cleaned => the tundish nozzle insufficiently cleaned.

l. 74: bubbles might realize argon blowing?? Is this so uncertain?

l. 82: Results cannot optimize anything!

l. 86: Do you really men that the model can operate, even though a slag layer affects it?

l. 90: a rigid sphere => rigid spheres

l. 102: here the k-epsilon?? k-epsilon what?

l. 111, 116, 119: write “where ….” instead if “Here, ..” after the equations

l. 106: Give reference to Launder and Spalding.

l. 109: What is velocity motion? Can you have a velocity without motion?

l. 113: Write D as subscript in C_D

l. 115, 118: It is not necessary to mention the name of an equation that follows immediately after the sentence.

l. 131: bubbles reflected => bubbles were reflected

l. 136: The schematic => A schematic

Table 1: 1e^-0.8 means e in the power of -8.

l. 157: by calculating 15 s of real time?

l. 159-: What is the “regional flow” here?

l. 164-165: Why change of tense? compared…analyze

l. 174: in this paper => in this work.

l. 186: bubble => bubbles

l. 190: the sentence “Thus the flow field…” is not justified by the present findings.

l. 195: This formulation sounds like the casting speed would automatically adjust the brick size: Please rephrase!

l. 204: liquid flow in the surface: How can a flow be “in” a surface?

l. 207: bubble => bubbles

l. 210: liquid level flow?

l. 213: What is “mathematical simulation calculation and theoretical analysis”?

l. 224: indicated that the argon => indicates that argon

l. 230: of stopper => of the stopper

l. 234: This then increased the flow rate of argon…?? In what sense “This”? Why did it increase the argon flow rate?

l. 235: the excessive => an excessive

l. 239: A useful argon flow.... should not be => The argon flow… should therefore not be

l. 243: increases and then increases? As a function of what?

l. 250: were removed and migrated? What/who removed them?

l. 252, 254: Minimize and maximize don’t mean the same as reduce and increase! Minimized distance would clearly here be 0 m.

l. 267: argon bubble => argon bubbles. high flow rate of argon may be appropriate. Why? 

l. 275: First mention the conditions (i.e., the matters in the parentheses), then the results (i.e., the first part of the sentence starting on l. 274).

l. 295: How can a position "decrease"?

l. 298-299: Thus, the high casting speed is disadvantage to removal of inclusions. => Thus, a high casting speed is disadvantageous for removal of inclusions.

l. 306: Do you really think such accurate numbers (with four digits) are useful or even appropriate? Besides, the volume fraction in “minimized” just because you didn’t include any cases with lower casting seed. Obviously, for a casting speed < 1 m/min, the argon fraction would be even lower!

Author Response

Point 1: The authors present a simulation study by CFD of the conditions in a tundish in steelmaking, with argon blowing. The authors simulate different argon feed rates and casting speeds and draw conclusions on whether the flow patterns are appropriate to remove inclusions and too yield desired flow patterns and residence time of the steel in the tundish. The simulated cases are clearly presented and the flow of the manuscript is good. Many results are reported, but it would be good to come up with more definite findings for the reader, i.e., to stress what can be learnt from the computational exercise. Some of the findings reported are rather obvious and could be given less emphasis (e.g., on l. 199 that molten steel entered the tundish from the shroud, and on l. 264 that at a higher flow rate of argon, more argon bubbles may be released from the annular permeable brick).  On the other hand, some quite definite findings are reported that are not justified, e.g., l. 215: Thus, the flow behavior of liquid steel in the tundish is reasonable (in what sense!?) when the annular argon blowing was adopted – the inclusions can float more easily (not demonstrated in any way in the analysis!).

Response 1: Thus, the flow behavior of liquid steel in the tundish is reasonable when the annular argon blowing was adopted—the inclusions can float more easily.” has been revised with “Thus, the annular argon blowing is beneficial to the floating and removal of inclusions in liquid steel.”

Point 2: The verification of the computational model is not appropriate. The authors’ section (3) on this topic merely holds simulation results and a few photographs. The bubble plume photo (Fig. 3b) does not allow for any further comparison with the simulated results. The CFD simulations in the latter figure have no scale, and to me the comparison gives practically nothing: the authors conclude that the verification indicates that the model is credible, but the material in section does not support the claim. Here, the authors have to be much more specific and also quantify their findings in some way.

Response 2: For the bubble plume photo (Fig. 3b) does not allow for any further comparison with the simulated results (Fig. 3a), it only demonstrates the floating behavior of argon bubbles in mathematical simulated results and water model is similar. Then, the subtitle 3 has been revised with “Comparison of flow behavior in mathematical model and water model”.

Point 3:  Fig. 4 the authors don’t even bother to say what the right figures represent.

Response 3: The left and right side in figure 4 has been explained.

Point 4: Throughout the text: Replace “liquid level” by “liquid surface”

Response 4: It has been corrected in paper.

Point 5: l. 12: model for  => model of

Response 5: It has been corrected

Point 6: l. 15: were consistent  => were found consistent

Response 6: It has been corrected

Point 7: l. 24: to avoid purity??

Response 7: Here, it means to avoid slag entrapment and to purify the molten steel.

Point 8: l. 30: Tundish metal => the tundish

Response 8:  It has been corrected.

Point 9: l. 33: the inclusion => inclusions

Response9: It has been corrected.

Point 10: l. 36: this nonmetallic inclusion => Nonmetallic inclusions

Response 10: It has been corrected.

Point 11: l. 43: on the => through the

Response 11: It has been corrected.

Point 12: l. 44: floatation => flotation

Response 12: It has been corrected.

Point 13: l. 46: Why mention both liquid and molten steel – are they different?

Response 13: Their meaning is the same, we have chosen the same description in paper, and “molten steel” has been replaced by “liquid steel”.

Point 14: l. 48: brick. This => brick, but

Response 14: It has been corrected.

Point 15: l. 50: entered into => entered

Response 15: It has been corrected.

Point 16: l. 51: argon blowed by stopper?

Response 16: In production, sometimes, the argon can be blown by the inner passage in stopper, and be injected into the liquid steel through the small hole in bottom end of stopper.

Point 17: l. 53: inclusion = inclusions

Response 17: It has been corrected.

Point 18: l. 60: et al => et al.

Response 18: It has been corrected.

Point 19: l. 66: the tundish nozzle unable to be sufficiently cleaned => the tundish nozzle insufficiently cleaned.

Response 19: It has been corrected.

Point 20: l. 74: bubbles might realize argon blowing?? Is this so uncertain?

Response 20: “might“ has been replaced with “can”.

Point 21: l. 82: Results cannot optimize anything!

Response 21: Results can be leveraged as a theoretical basis for the optimization of the annular argon blowing at upper nozzle and the improvement of slab quality. Then it has been revised in paper.

Point 22: l. 86: Do you really ment that the model can operate, even though a slag layer affects it?

Response 22: The description is not favorable. The flow of liquid steel is focus of attention, so we have not considered the slag layer. The assumption (1) has been replaced: “Regardless of the influence of the liquid slag on the flow behavior of the molten steel.”

Point 23: l. 90: a rigid sphere => rigid spheres

Response 23: It has been corrected.

Point 24: l. 102: here the k-epsilon?? k-epsilon what?

Response 24: The description is not favorable, it has been corrected.

Point 25: l. 111, 116, 119: write “where ….” instead if “Here, ..” after the equations

Response 25: It has been corrected.

Point 26: l. 106: Give reference to Launder and Spalding.

Response 26: The reference has been added.

Point 27: l. 109: What is velocity motion? Can you have a velocity without motion?

Response 27: motion has been deleted.

Point 28: l. 113: Write D as subscript in C_D

Response 28: It has been corrected.

Point 29: l. 115, 118: It is not necessary to mention the name of an equation that follows immediately after the sentence.

Response 29: It has been corrected.

Point 30: l. 131: bubbles reflected => bubbles were reflected

Response 30: It has been corrected.

Point 31: l. 136: The schematic => A schematic

Response 31: It has been corrected.

Table 32: 1e^-0.8 means e in the power of -8.

Response 32: It has been corrected.

Point 33: l. 157: by calculating 15 s of real time?

Response 33: The time is argon blowing time. When the argon blowing time was chose with 15 s in calculation, the flow field in tundish was stable.

Point 34: l. 159-: What is the “regional flow” here?

Response 34: The description is not favorable, it has been corrected. “the tracer with the same properties as the regional fluid was added” has been corrected with ” the tracer with the same properties as the liquid steel was added

Point 35: l. 164-165: Why change of tense? compared…analyze

Next, we compared the vertical velocity of the liquid steel in the vicinity of stopper to analyze the influence of different process parameters on the flow of liquid steel.

Response 35: There is a “to” before “analyze”.

Point 36: l. 174: in this paper => in this work.

Response 36: It has been corrected.

Point 37: l. 186: bubble => bubbles

Response 37: It has been corrected.

Point 38: l. 190: the sentence “Thus the flow field…” is not justified by the present findings.

Response 38: This sentences has been deleted.

Point 39: l. 195: This formulation sounds like the casting speed would automatically adjust the brick size: Please rephrase!

Response 39: It has been corrected.

Point 40: l. 204: liquid flow in the surface: How can a flow be “in” a surface?

Response 40: It has been corrected.

Point 41: l. 207: bubble => bubbles

Response 41: It has been corrected.

Point 42: l. 210: liquid level flow?

Response 42: It has been corrected.

Point 43: l. 213: What is “mathematical simulation calculation and theoretical analysis”?

Response 43: It has been corrected.

Point 44: l. 224: indicated that the argon => indicates that argon

Response 44: It has been corrected.

Point 45: l. 230: of stopper => of the stopper

Response 45: It has been corrected.

Point 46: l. 234: This then increased the flow rate of argon…?? In what sense “This”? Why did it increase the argon flow rate?

Response 46: It has been revised with “, and then the increased flow rate of argon may improve the purity of liquid steel.

Point 47: l. 235: the excessive => an excessive

Response 47: It has been corrected.

Point 48: l. 239: A useful argon flow.... should not be => The argon flow… should therefore not be

Response 48: “A useful argon flow rate should not be larger than 3 L/min.” has been revised “The argon flow rate should therefore not be larger than 3 L/min.

Point 49: l. 243: increases and then increases? As a function of what?

Response 49: “As the distance from the upper nozzle surface increases,”  it has been added in paper.

Point 50: l. 250: were removed and migrated? What/who removed them?

Response 50: “and some argon bubbles were removed and migrated to the right side of the stopper” has been revised with “and some argon bubbles were carried by the liquid steel to the right side of the stopper”

Point 51: l. 252, 254: Minimize and maximize don’t mean the same as reduce and increase! Minimized distance would clearly here be 0 m.

Response 51: “when the distance between the front wall of the tundish and stopper is minimized, the floating argon bubble cannot easily be dispersed. The driving effect of the floating bubbles then increases, and the vertical flow velocity of the liquid steel in the front vicinity of the stopper is maximized.” Has been revised as “ the distance between the front wall of the tundish and stopper is smaller, the floating argon bubble cannot easily be dispersed. The driving effect of the floating bubbles then increases, and the vertical flow velocity of the liquid steel in the front vicinity of the stopper is higher.”

Point 52: l. 267: argon bubble => argon bubbles. high flow rate of argon may be appropriate. Why? 

Response 52: The high flow rate of argon blowing can carry more argon bubbles into the nozzle, which can prevent the nozzle from clogging, moreover, more argon bubbles can be generated and float  up in tundish, which is favorable the remove of inclusions.

Point 53: l. 275: First mention the conditions (i.e., the matters in the parentheses), then the results (i.e., the first part of the sentence starting on l. 274).

Response 53: It has been corrected in paper.

Point 54: l. 295: How can a position "decrease"?

Response 54: “the height of peak value position also decreases,”     and it has been corrected in paper.

Point 55: l. 298-299: Thus, the high casting speed is disadvantage to removal of inclusions. => Thus, a high casting speed is disadvantageous for removal of inclusions.

Response 55: it has been corrected in paper.

Point 56: l. 306: Do you really think such accurate numbers (with four digits) are useful or even appropriate? Besides, the volume fraction in “minimized” just because you didn’t include any cases with lower casting seed. Obviously, for a casting speed < 1 m/min, the argon fraction would be even lower!

Response 56: The number of four digits has been revised with three digits.

This sentence has shown that at low casting speed, the volume ratio of argon entering the nozzle is low, and the volume ratio of argon entering the nozzle is high at high casting speed. 

Reviewer 2 Report

On line 91, The Rosin-Rammler distribution for bubble sizes was chosen? Please, justify why, on which basis.

On line 94, "...three-dimensional transient incompressible flow" Yet, in eq. 2 the unsteady term is missing.

On line 98, Continuity equation is totally wrong! Divergence of divergence...?

On line 100, "ui and uj are the velocity components along the i and j directions respectively"

This is misleading, one could think that the author has the coordinate system i,j,k in mind! Which would be wrong. In eq. 2, Subscripts apply to the Einstein summation technique.

On line 117, Eq. 9 is not a species transport equation. It is a concentration advection-diffusion equation.

On line 119, "u,v,w is the three-dimensional velocity flux of molten steel"

I would not call it a flux at the first place, but more importantly the author should use the same symbols as in eq. 2, i.e. the velocity vector is ui and not u,v,w.

On line 119, unit of diffusivity coefficient is incorrect! It should be m2/s

On lines 127-128, "...was set as a free surface" This is very unclear definition of boundary condition, especially when ANSYS Fluent is concerned. Please, explain, which boundary condition was applied at the free surface.

On line 129, on symmetry wall, the boundary condition for velocity field is a zero normal component and zero gradient of tangential velocity component. Please, correct in the text.

Online 131, "...with standard wall functions" Standard wall functions typically suffers from an excessive numerical diffusion. There are better choices than that.

On line 141, diffusivity coefficient unit again wrong! it should be m2/s.

On line 150, "grid refinement" is not used to simulate anything! Try to come up with a more appropriate explanation of why the grid refinement was used.

On line 155, I wonder how the termination criteria was chosen. Why 1e-4?

On the same line, how was the size of the time step determined?

On line 171, "...affected by the viscous force, gravity, and inertial forces" By saying this you name all the forces from the N-S eqs; therefore, such information is absolutely redundant and unnecessary.

On lines 189-191, A visual observation of bunch of particles can hardly be used to claim that the model is credible. Please, agree with me that quantitative data is required to do so.

On lines 195, 3D path lines are nice on a PC's screen, but significantly less useful on a sheet of paper. In scientific articles, such figures should be avoided.

On lines 264-267, Two sentences say the same thing. A redundant text should be avoided to easy reader's reading.

On line 267, "...can prevent the nozzle from clogging" However, clogging was not an objective of the numerical efforts. How can it be claimed then?

On line 315, "...molten steel in the tundish slightly decreases." According to table 3, It however increases.

To conclude, In introduction, the authors say: "The results can optimize the annular argon blowing at the upper nozzle and improve slab quality."

Unfortunately, in the whole manuscript, there is no word about optimizing the argon flow in a tundish or any recommendation on setting of parameters. Therefore, it is also not clear to me how the slab quality can be improved.

Author Response

Point 1:  On line 91, The Rosin-Rammler distribution for bubble sizes was chosen? Please, justify why, on which basis.

Response 1: The bubble diameters are distributed by Rosin-Rammler statistics, which were obtained by the water model experiments of tundish.

Point 2:  On line 94, "...three-dimensional transient incompressible flow" Yet, in eq. 2 the unsteady term is missing. On line 98, Continuity equation is totally wrong! Divergence of divergence...?

Response 2: The equation 1, 2, 3, 4 and 5 have been corrected.

Point 3:  On line 100, "ui and uj are the velocity components along the i and j directions respectively" This is misleading, one could think that the author has the coordinate system i,j,k in mind! Which would be wrong. In eq. 2, Subscripts apply to the Einstein summation technique.

Response 3:  The explain of subscripts i and j in eq.2 has been revised.

Point 4:  On line 117, Eq. 9 is not a species transport equation. It is a concentration advection-diffusion equation.

Response 4: Eq. 9 is a concentration advection-diffusion equation, and is used to describe the time evolution of the species mass fraction C in tundish. The meaning has been revised in paper.

Point 5:  On line 119, "u,v,w is the three-dimensional velocity flux of molten steel".I would not call it a flux at the first place, but more importantly the author should use the same symbols as in eq. 2, i.e. the velocity vector is ui and not u,v,w.

Response 5: The equation 9 has been revised.

Point 6:  On line 119, is incorrect! It should be m2/s

Response 6: The error unit of diffusivity coefficient has been corrected.

Point 7:  On lines 127-128, "...was set as a free surface" This is very unclear definition of boundary condition, especially when ANSYS Fluent is concerned. Please, explain, which boundary condition was applied at the free surface.

Response 7: The explanation has been added in line 128.

Point 8:  On line 129, on symmetry wall, the boundary condition for velocity field is a zero normal component and zero gradient of tangential velocity component. Please, correct in the text.

Response 8: It has been corrected.

Point 9:  Online 131, "...with standard wall functions" Standard wall functions typically suffers from an excessive numerical diffusion. There are better choices than that.

Response 9: The volume of tundish is larger, the effect of boundary layer on the flow of liquid steel is smaller, and moreover, the molten steel flow in tundish is in the range of high Reynolds number. The standard wall function can well satisfy the calculation precision of high Reynolds number mathematical models. So, we chose the standard wall functions.

Point 10:  On line 141, diffusivity coefficient unit again wrong! it should be m2/s.

Response 10:  It has been corrected.

Point 11:  On line 150, "grid refinement" is not used to simulate anything! Try to come up with a more appropriate explanation of why the grid refinement was used.

Response 11: The velocity of liquid steel at the inlet and outlet of tundish is larger, and the flow behavior in the vicinity of stopper should be paid more attention. Then the grid refinement should be conducted in the vicinity of stopper, inlet and outlet of tundish. The corresponding explanation has been added in paper.

Point 12:  On line 155, I wonder how the termination criteria was chosen. Why 1e-4? On the same line, how was the size of the time step determined?

Response 12: The calculation was considered to be converged when the normalized residuals of all variables were smaller than 10-4, when the flow field could achieve a relatively stable state. To reduce the simulation time and ensure the balance of equations at discrete point of time, a fixed time step of 0.005 s was used in the time-dependent solution.

Point 13:  On line 171, "...affected by the viscous force, gravity, and inertial forces" By saying this you name all the forces from the N-S eqs; therefore, such information is absolutely redundant and unnecessary.

Response 13: In order to ensure the similarity of the water model and prototype model, the Froude number should be equal in the water model and prototype model, the Froude number is related the viscous force, gravity, and inertial forces.

Point 14:  On lines 189-191, A visual observation of bunch of particles can hardly be used to claim that the model is credible. Please, agree with me that quantitative data is required to do so.

Response 14: The reviewer is correct. however, it can provide the qualitative analysis, so the subtitle 3 has been revised with “Comparison of flow behavior in mathematical model and water model” 

Point 15:  On lines 195, 3D path lines are nice on a PC's screen, but significantly less useful on a sheet of paper. In scientific articles, such figures should be avoided.

Response 15: The velocity streamlines can show that the flow path of liquid steel, the difference of flow behavior of liquid steel with argon blowing and without argon blowing can be distinguished.

Point 16:  On lines 264-267, Two sentences say the same thing. A redundant text should be avoided to easy reader's reading.

Response 16: The next sentence has been deleted.

Point 17:  On line 267, "...can prevent the nozzle from clogging" However, clogging was not an objective of the numerical efforts. How can it be claimed then?

Response 17: For the argon blowing in the tundish nozzle is beneficial to the prevention of nozzle clogging, the appropriate argon blowing by the annular argon blowing brick can increase the argon volume in the tundish nozzle, so it can be conclude that “...can prevent the nozzle from clogging”.

Point 18:  On line 315, "...molten steel in the tundish slightly decreases." According to table 3, It however increases.

Response 18: The mistake has been corrected.

Point 19:  To conclude, In introduction, the authors say: "The results can optimize the annular argon blowing at the upper nozzle and improve slab quality."

Response 19: This statement is not suitable, so this sentence has been revised with “The results can be leveraged as a theoretical basis for the optimization of the annular argon blowing at upper nozzle and the improvement of slab quality.”

Point 20:  Unfortunately, in the whole manuscript, there is no word about optimizing the argon flow in a tundish or any recommendation on setting of parameters. Therefore, it is also not clear to me how the slab quality can be improved.

Response 20: The results can be leveraged as a theoretical basis for the optimization of the annular argon blowing at upper nozzle and the improvement of slab quality. At the same time, we have discussed the reasonable flow rate of argon blowing combined with the surface flow velocity of liquid steel in 4.2, and believed that the argon flow rate should therefore not be larger than 3 L/min.

Round  2

Reviewer 1 Report

The authors have done most of the changes suggested in the review, but some small matters still remain.

l. 87: Item (1) should be rewritten, e.g., to “The effect of liquid slag on the flow behavior of liquid steel is neglected.”

l. 97: equation, the momentum  => equation and the momentum

l. 100, 105, 112 and 120: Sentences where an equation appears inside the sentence should not end in mentioning the Equation number (it is useless) and a colon (:) before the equation, and should not be followed by an indented “Where, a is…”.  This would mean that the “Where..” sentence is a new sentence, which it is not, and the comma after where is really odd! For example, the sentence with Eq. (8) should be written as (where I don’t write the symbols correctly here)

“ …divided into discrete groups as shown in

Y_d = …

where Y_d is the …”

l. 103-104: Leave a space before (k) and (epsilon)

l. 153: more attention. => more attention to.

l. 282 and elsewhere in the manuscript: Always leave a space before the unit; write 220 mm and not 220mm.

Author Response

Point 1: l. 87: Item (1) should be rewritten, e.g., to “The effect of liquid slag on the flow behavior of liquid steel is neglected.”

Response 1: The effect of liquid slag on the flow behavior of liquid steel in tundish is neglected

Point 2: l. 97: equation, the momentum  => equation and the momentum

Response 2: It has been corrected in paper.

Point 3: l. 100, 105, 112 and 120: Sentences where an equation appears inside the sentence should not end in mentioning the Equation number (it is useless) and a colon (:) before the equation, and should not be followed by an indented “Where, a is…”.  This would mean that the “Where..” sentence is a new sentence, which it is not, and the comma after where is really odd! For example, the sentence with Eq. (8) should be written as (where I don’t write the symbols correctly here)

“ …divided into discrete groups as shown in

Y_d = …

where Y_d is the …”

Response 3: It has been corrected in paper.

Point 4: l. 103-104: Leave a space before (k) and (epsilon)

Response 4: It has been corrected in paper.

Point 5: l. 153: more attention. => more attention to.

Response 5: It has been corrected in paper.

Point 6: l. 282 and elsewhere in the manuscript: Always leave a space before the unit; write 220 mm and not 220mm.

Response 6: It has been corrected in paper.

Reviewer 2 Report

Dear Authors,

I appreciate your efforts in writing the manuscript. I must agree you tried to address all the points of myself as a review. I did quite a good job. However, I must frankly admit as a reader I am not satisfied with the way you present your results.

It is really hard to get a main message of the article and pasting snapshots of contours and pathlines from Fluent makes things even worse.

Author Response

Point 1: l. I appreciate your efforts in writing the manuscript. I must agree you tried to address all the points of myself as a review. I did quite a good job. However, I must frankly admit as a reader I am not satisfied with the way you present your results.

It is really hard to get a main message of the article and pasting snapshots of contours and pathlines from Fluent makes things even worse.

Response 1: Thank you very much for the reviewer's suggestion. We have revised the Figure 5, 6 and 9 in the paper. The clarity extent of figure 5 has been improved. The velocity vector diagram of molten steel in tundish in Fig. 6 and 9 are redraw by the interpolation method of Tecplot software by using of the data from the Fluent software
